# Induced interpretation bias affects free recall and episodic memory bias in social anxiety

**Hye Ryeong Park, Jong-Sun Lee** *

Department of Psychology, Kangwon National University, Chuncheon, Korea

* jongsunlee@kangwon.ac.kr

**Data Availability Statement:** All relevant data are within the paper and its Supporting Information files.

**Funding:** This work was supported by the National Research Foundation of Korea (NRF) grant funded

## Abstract

The combined effect of each cognitive bias, interpretation, attention, and memory bias, is known to play a causal role in the etiology and maintenance of social anxiety. However, little is known about how each type of bias (i.e., interpretation, memory bias) acts during social anxiety. The present study aimed to investigate whether experimentally induced interpretation bias using the cognitive bias modification (CBM) paradigm would influence free recall and episodic memory biases in a Korean sample. A total of 61 participants were randomly assigned to either a positive (n = 30) or negative (n = 31) CBM group. The study used CBM scenarios that were auditory-specific and focused on social anxiety symptoms. The results showed that interpretation biases could be induced, and they resulted in training congruent state mood and memory biases on both free-recall memory and autobiographical memory, which partly confirmed the combined cognitive biases hypothesis proposed by Hirsch, Clark (1).

## Introduction

According to the 2016 Psychological Disease Dynamics Survey released by the Ministry of Health and Welfare of Korea, the demographic distribution of the annual prevalence rate of social anxiety disorder in 2016 was 1.0% in the age group of 18–29 years, and this figure is about twice as high as the below 0.5% for other age groups [1]. In a prior study on Korean university students, 54.9% of participants experienced more than moderate anxiety at job interviews and 8.5% in daily encounters with other people [2]. As such, social anxiety disorder as found in people in their 20s is a painful disorder that directly affects a person's daily life, impeding their academic, social, and other functional aspects and tends to persist chronically without treatment [3].

Cognitive biases such as interpretation and memory bias has been known to be significant and potent risk factors to cause and maintain social anxiety. Many prior studies have shown that people with social anxiety interpret ambiguous social information as a threat [4,5]. For example, Amir, Foa and Coles [5] presented ambiguous social or non-social situations to three groups; clinical groups of people with generalized social phobia and obsessive-compulsive disorder, as well as a control group without social anxiety. The results showed that the socially anxious group tended to interpret ambiguous social scenarios more negatively compared with the other groups such as obsessive-compulsive group and the control group; however, such

by the Korea government (MSIT) (No. 2018R1C1B5043272). There was no additional external or internal funding received for this study.

**Competing interests:** The authors have declared that no competing interests exist.

tendency was not found in non-social scenarios. Similarly, Stopa and Clark [4] observed that the socially anxious group negatively interpreted ambiguous social situations and slightly negative social events as catastrophic, compared with the other groups with different anxiety disorders (panic disorder, simple phobia, agoraphobia, post-traumatic stress disorder) or healthy control. Furthermore, prior studies have argued that people with social anxiety may lack a positive bias [6,7]. Indeed, the social anxiety group showed lack of positive interpretations of ambiguous social situations, compared with healthy control [7]. Thus, there is a need for an intervention that corrects negative interpretation bias and promotes positive interpretation bias to effectively treat anxiety and/or depression.

Cognitive Bias Modification of interpretation (CBM-I) aims to positively correct negative biases individuals with anxiety or depression, by providing repetitive computer-based tasks for interpreting ambiguous social situations in a more flexible and positive way [8]. Participants can complete the training task at home or wherever they prefer. Mathews and Mackintosh [9] argued that CBM-I, which induces positive interpretations in social anxiety, may have therapeutic implications that reduce anxiety caused by negative interpretation bias. In support this, Mathews and Mackintosh [9] found that participants in the positive CBM-I group showed significantly reduced levels of anxiety as well as increased positive interpretation bias compared with participants in the negative CBM-I group, confirming the causal relationships between cognitive bias and mood. Subsequent studies with the subclinical population with social anxiety symptoms have since demonstrated iterative CBM-I training can modify interpretation bias for social situations and then affect the change of symptoms [8,10,11]. Previous studies have explored the optimal mode of delivering CBM-I intervention that involve short stories based on daily life experiences. It has been hypothesized that imagining scenarios mentally is more effective in influencing mood compared to verbal reasoning. Indeed, several studies have demonstrated that auditory CBM-I that utilizes imagery has a greater impact on mood and interpretation biases compared to visually presented CBM-I intervention [12–14].

Memory bias is another factor that prolongs social anxiety. Individual with high levels of social anxiety showed the tendency to recall negative or threatening memories more easily than positive ones in social situations [15]. However, studies looking at memory bias in social anxiety disorder have shown conflicting results. Rapee and colleagues [16] reported that social anxiety and non-clinical control groups showed no significant difference in recalling socially threatening, physically threatening, and neutral and positive words. Becker and colleagues [17] did not find also negative memory bias in participants with social anxiety. Meanwhile, in a study by Cody and Teachman [15], participants delivered speeches and performed recognition tasks after hearing feedbacks for their own and others' speeches. Participants with social anxiety remembered the feedback other participants received positively but remembered their own feedback negatively. Thus, they showed negative memory bias with regard to their own evaluation. Glazier and Alden [18] gave participants positive or neutral feedback after their speeches and investigated the changes by measuring memory bias after five minutes and after one week, respectively. The results showed that a higher level of social anxiety was related to a higher tendency to remember positive feedback negatively as time passes. As with the preceding studies above, the results of studies on memory bias in social anxiety disorders are inconsistent and need to be further studied.

On the other hand, Hirsch and Clark [19] argued that cognitive biases will influence and interact with one another in maintaining social anxiety disorders, setting out the combined cognitive biases hypothesis. Based on this hypothesis, few studies have been conducted recently to verify the association between cognitive biases in social anxiety disorders. For example, Hertel and colleagues [20] investigated the association between interpretation and memory biases in social anxiety group and control groups. Participants were asked to interpret

ambiguous but potentially threatening social scenarios and then to remember the scenarios. Upon recollection of the scenarios, the social anxiety groups reported a significantly higher percentage of false errors containing emotionally negative content when compared to the control group. Tran, Hertel and Joormann [21] recruited university students to examine the impact of positive or negative interpretation bias on memory bias using the single session of CBM-I training. One group of participants was trained to interpret ambiguous social scenarios positively, and the other, negatively. Subsequently, the researchers measured the interpretation and memory biases to see if there would be interpretation training-congruent memory biases. Results showed that the group trained to interpret positively recalled more positive memories compared with the group trained to interpret negatively whereas the negative interpretation training revealed opposite results, showing a bias that matched the training affect. These results partially supported the combined cognitive biases hypothesis of Hirsch and Clark [19] that interpretation bias affects memory. These prior studies suggest that modifying interpretation bias for events will affect subsequent memory and can also modify memory bias.

Summing up the results of prior studies mentioned above, CBM-I can affect memory bias as well as interpretation bias. However, not many studies have examined the interaction between interpretation and memory bias using CBM-I training. Holmes and Mathews [22] argued that the auditory method of reading out the scenario and having participants imagine it has a greater impact on mood change compared with the verbal processing method. Meanwhile, a previous study on the relationship between interpretation and memory biases have been conducted in a visual manner in which verbal processing is likely to occur [21]. Tran, Hertel and Joormann [21], in examining the effects of CBM-I on interpretation and memory bias, conducted a mood rating at three times (before CBM-I training, after CBM-I training, and after recall task) for the positive CBM-I group and negative CBM-I group, which did not show significant main effects of group and time and interaction effect of time × group. This result may be due to the CBM-I training were not conducted in auditory mode as in Holmes and Mathews [22]'s study. Therefore, the present study intends to conduct the auditory version of CBM-I training to prevent verbal processing and further to influence the state mood. Unlike Tran, Hertel and Joormann [21], which used comprehension questions to emphasize scenario in CBM-I, we use comprehension questions that emphasize the five senses to help imagery. Also Tran, Hertel and Joormann [21] presented a CBM-I scenario consisting of general social situations rather than those directly related to social anxiety symptoms specific social situation; thus, their work has the limitation of not presenting scenarios by which one could examine social anxiety levels or changes in symptoms. Therefore, the present study aimed to supplement the work of Tran, Hertel and Joormann [21] in a preliminary study by creating a CBM-I training program with an auditory scenario related to social anxiety and then examining its effectiveness. Specifically, after conducting a CBM-I training program using positive images and a CBM-I training program using negative images on healthy university students, we examined the changes in state mood, interpretation bias, and both free-recall and episodic memory. Furthermore, the previous studies included anyone for participants without a screening process; they presented only mean scores of depression or anxiety symptoms to show the homogeneity of positive and negative CBM conditions. The reason for this finding is that participants in the present study, unlike in prior studies, are healthy people without social anxiety and depression. In previous works [9,21], the participants were likely to be recruited without a screening process and so included a mix of healthy people as well as those who tend to be depressed and socially anxious. As the present study selected only healthy participants who have neither social anxiety nor depression through a screening test, they might have had a greater difficulty in having negative mental images.

Without limiting cognitive bias to interpretation bias, we examined the effects of CBM-I on interpretation and memory biases. The research hypotheses are as follows:

Hypothesis 1. The positive CBM-I training group will show increased happy and decreased sad mood whereas the negative CBM-I training group will show decreased happy and increased sad mood.

Hypothesis 2. The positive CBM-I training group will have more positive interpretations than negative ones on neutral scenarios, whereas the negative CBM-I training group will have more negative interpretations than positive ones.

Hypothesis 3. The positive CBM-I training group will recall more positive memories than neutral or negative ones after training, whereas the negative CBM-I training group will recall more negative memories than neutral or positive ones.

## Method

### Participants

A total of 114 undergraduate and graduate students completed an online screening survey using the Center for Epidemiological Studies-Depression(CES-D), Social Avoidance Distress Scale (SADS), and Brief version of Fear of Negative Evaluation Scale (B-FNE). Participants who scored under 16 points on the CES-D, $< 99$ points on the SADS, and $< 48$ points on the B-FNE ($n = 65$) and voluntarily agreed to participate were invited to be a part of the experiment ($n = 61$). On the day of the experiment, the participants completed the written consent form followed by the secondary screening survey. If a subject met the research criteria, they were randomly assigned to either a positive CBM-I group ($n = 30$) or a negative CBM-I group ($n = 31$). However, the data of one participant who backed out while receiving training were excluded from the final analysis, leaving a total of 30 participants in the negative CBM-I group.

### Questionnaires

**Center for Epidemiological Studies-Depression (CES-D).** The CES-D [23,24] was used during the screening phase to measure the degree of depression experienced in the previous week, using a four-point Likert scale (0 = extremely rare to 3 = most likely). It has demonstrated strong psychometric properties, including a high internal consistency coefficient (Radloff [23] Cronbach's α = .85; Cho and Kim [24] Cronbach's α = .91) and satisfactory test-retest reliability (Radloff [23] r = .57; Cho and Kim [24] r = .68). Our study yielded a Cronbach's α of .70.

**Social Avoidance and Distress Scale (SADS).** The SADS [25,26] was used in the screening phase to evaluate the anxiety and inclination of participants to avoid social situations on a five-point Likert scale (ranging from 1 = do not agree at all to 5 = strongly agree). It has good psychometric properties, including a high internal consistency coefficient (Lee and Choi [26] Cronbach's α = .92) and satisfactory test-retest reliability (Watson and Friend [25] r = .68; Lee and Choi [26] r = .88). Our Cronbach's α was .88.

**Brief Version of Fear of Negative Evaluation Scale (B-FNE).** The B-FNE [25–27] was used during the screening phase to assess the fear of being negatively evaluated by others, which is a key characteristic of social anxiety. Response were given on a five-point Likert scale ranging from (*not at all*) to 5 (*very much*). The questionnaire has been reported to have good psychometric properties, including high internal consistency coefficient (Lee and Choi [26]; Leary Lee and Choi (26), [27] Cronbach's α = .90) and good test-retest reliability (Watson and Friend [25] *r* = .78; Lee and Choi [26] *r* = .80; Leary [27] *r* = .75). Our study obtained a Cronbach's *α of* .89.

**Visual Analogue Scale (VAS).** To assess participants' state mood, they were asked to rate their mood before and after the training and the first filler task. The VAS was used as a self-

report measure of current emotions and pain. The VAS is a simple tool that allows individuals to rate their subjective state [28] by marking an 'X' on a horizontal bar ranging from 0 (not at all) to 10 (extremely) for negative (sadness) and positive (happiness) emotions. This scale is ideal for self-reporting as the response burden is low [29].

**Six-item short version of the state-trait anxiety inventory (STAI-6).** STAI-6 [30] is the shorter version of the State-Trait Anxiety Inventory-State (STAI-S) [31]. It assesses the current level of anxiety using a four-point Likert scale (1 = *not at all* to 4 = *strongly agree*). Compared to STAI-S, it is a concise, reliable, and valid measure that is sensitive to changes in anxiety [30,32]. Previous studies have reported high internal consistency coefficient (Tluczek, Henriques [32] Cronbach's $\alpha$ ranging from .74 to .82). In our study, the Cronbach's $\alpha$ was .74 before training, .80 after training, and .77 after the filler tasks.

## Training

**Imagery training.** Imagery practice of approximately 10 minutes was conducted for the participants before the CBM-I training. Imagery practice was based on Oxford Imagery Generation (OxIGen) used by Blackwell and colleagues [12] and Pictet, Jermann [33]. Participants were asked to close their eyes and imagine the objects or scenarios mentioned using all five senses. Participants were asked to imagine that they were the main character of a story unfolding in the present. The researcher asked questions invoking the use of five senses (for example, can you smell the fragrance from the lemon?) so that the participant could focus on the imagination. Participants rated the vividness on a nine-point Likert scale (1 = not at all vivid to 9 = extremely vivid) and participated in CBM-I training when their score exceeded seven-points.

**Cognitive Bias Modification-Interpretation (CBM-I).** CBM-I was structured on interpretation training with imagery used by previous studies [9,14,22]. Holmes and Mathews [22] made participants imagine the scripts that always ended positively or negatively using auditory methods (e.g., "I suddenly sneezed very loudly during class. Everyone is looking towards me. To people, my behavior is. . . . . .. *not a big deal* (positive condition)/*a big deal* (negative condition)., followed by a comprehension question emphasizing the use of five senses (positive condition: "Do you see the people who think my sneeze is not a big deal?"; negative condition: "Do you see the people who think my sneeze is distracting?"). The participants were then asked to press the Yes (Y) or No (N) button using a computer keyboard, followed by correct or incorrect feedback. While Holmes and Mathews [22] presented comprehension questions on a computer screen, the present study used an auditory method so that the participants could focus on the imagery itself. If the participant imagined and responded to the given condition, the correct answer was presented on a computer screen. The participants were then asked to repeat the previously presented imagined scenarios aloud.

Salemink, Kindt [34] indicated that constructing the content of a CBM-I scenario that matches the interests of the participants would increase the training effect. Based on this feedback, the present study modified CBM-I content that relates to social situations commonly experienced by college students inside and outside campus (for example, oral presentation, part-time job, etc.). In addition, scenarios included content related to social anxiety symptoms. For example, in the last sentence of the scenarios for the CBM-I positive group, we specifically mentioned symptoms of social anxiety "my breathing is relaxed while introducing myself" instead of just saying "I complete self-introduction well." Five practice trials were conducted before the main CBM-I training session. Sixty trials were conducted (20 trials per block, with a total of three blocks). Scenarios within the blocks were randomly presented. After each block, participants evaluated how vivid and positive (or negative) imagined scenarios were on a scale

of 1 (not at all) to 7 (very). The training was generated using E-prime 2.0, and all scenarios were recorded in an identical female voice. Each scenario lasted approximately 17 to 21 seconds and the overall training took approximately 50–60 min.

### Test phase

**Similarity Rating Task (SRT).** The SRT was used in this study to assess the participants' interpretation bias after CBM training. The procedure and format of the SRT were based on the previous studies [35,36]. The SRT comprises two phases: encoding and recognition. In the encoding phase, below the title of each story, participants were presented with 10 scenarios, each with an ambiguous and neutral descriptions ending in a word fragment completion, followed by a neutral comprehension question ('yes or no' answer) regarding the content of each description. An example is as follows.

<Presentation of liberal arts class>

You give a presentation in a liberal arts class.

After the presentation, some people ask you questions.

As you answer the last question, the classroom goes sil__nt.

Did you answer the question?

[yes or no]

In the following recognition phase, each scenario's title was presented along with four types of sentences in random order: (1) positive sentence related to the scenario (target positive; i.e., *people are happy with my answer*), (2) negative sentence related to the scenario (target negative; i.e., *people are so frustrated with my answer*), (3) positive sentence unrelated to the scenario (foil positive; i.e., *you finish your presentation on time.*), and (4) negative sentence unrelated to the scenario (foil negative; i.e., *you don't finish your presentation on time*). Participants were asked to rate how similar each sentence was to the scenario they had read in the encoding phase on a four-point Likert scale (1 = very different; 2 = fairly different; 3 = fairly similar; 4 = very similar). The target sentences were either positive or negative interpretation related to the social situation scenario, the foil sentences were generally positive or negative [37]. The four types of sentences were fixed for each scenario and were presented in a random order. This study included 40 trials, with each scenario having four different trials consisting of a target positive, a target negative, a foil positive, and a foil negative interpretation.

The SRT was conducted using the E-prime 2.0 software and displayed on a 16-inch laptop screen with a white background and navy-colored Arial font. Before actual test, participants completed five practice trials. The mean score for each type of the sentence (target positive, target negative, foil positive, and foil negative) was calculated and used in the final analysis.

**Free recall task.** To assess memory bias related to the scenarios, the free recall task used by Joormann, Waugh [38] and Park and Lee [39] was included in the study. When the title of the scenario shared during the recognition task was presented one by one on the laptop screen, the participant was asked to recall the contents of the scenario and write them on a laptop placed next to them. There was no time limit; when the participant pressed a space bar after typing, the title of the next scenario was presented randomly.

The coding format used by Joormann, Waugh [38] and Park and Lee [39], Salemink, Heartel, and Makintosh [40] was also used in the present study. If there was an intrusion (reporting of new content that did not apply to the scenarios or recognition task sentences) in the recall of the participant, the valence of intrusion was coded as positive, negative, or neutral. In addition, new ideas (reporting of autobiographical memory), errors (case of confusion with a different scenario or sentence), and missing components (not completed or reported as not remembering) were coded. We also coded the valence of all the content recalled by

participants as positive, negative, or neutral. For example, a statement such as "I was nervous and did my presentation awkwardly, but I managed to complete it well and received good feedback" which included a negative sentence but had overall positive content, was coded as positive. Coding was conducted by two graduate students who were blinded to the CBM-I allocation. For inter-coder reliability, each of the two coders coded data from 20 subjects, which constituted more than 30% of the total participants. The inter-coder reliability for the inclusion of content was $\kappa = .98$, and the content valence was more than .90 for positive, negative, and neutral content. The reliability of the inclusion of intrusion was $\kappa = .94$, and the intrusion valence was more than .86. The reliability of the inclusion of new ideas was $\kappa = 1.00$, and the errors and missing components were more than .90.

**Past recollection task.**   To examine whether there was a bias in not only recalling the scenario but also a memory from the participant's actual past, the future thinking task used by MacLeod and Byrne [41] was slightly modified to examine the memory of the past. The participants were asked to narrate as many positive, negative, or neutral events as possible from three time periods ("the past week," "the past year," "the past 10 years") in one minute each. The events reported by the participants were recorded on a computer. As in previous studies [42], the total score is the total number of positive, negative, and neutral events mentioned by the participant in the three minutes, and if one event was repeated at different time points, only the first time was counted. Two graduate students who did not know the purpose of the study and the CBM-I allocation coded the events reported by the participants as positive, negative, or neutral. For inter-coder reliability, each of the two coders coded data collected from 20 subjects, which constituted more than 30% of the total participants. Intraclass correlation coefficients (ICCs) were used to analyze inter-coder reliability. The results of the analysis showed high reliability between the coders (ICC = .89, $p < .001$, 95% CI [.73, .96]).

## Procedure

The present study was approved by the authors' University's Institutional Review Board (KWNUIRB-2017-05-006-003). Participants were recruited via internet communities and university bulletin boards. When the participants contacted the researchers, they received a preliminary screening survey link to complete. If the participant met the participation criteria, the participant visited the laboratory to complete the secondary screening survey. Finally, if they met the participant criteria, a description of the study was shared and consent was requested. After the participants shared their consent, the experiment was conducted (Fig 1).

Participants were randomly assigned to either a positive CBM-I group or a negative CBM-I group. They completed the VAS (happiness, sadness) for baseline measurement and participated in approximately 10 min of imagery training, followed by CBM-I training. Subsequently, the VAS and STAI-6 was completed. To minimize the effect of training-based state mood changes on the SRT, participants participated in the first filler task (reading a neutral story) for approximately five minutes. The participants then completed the VAS and STAI-6 once again followed by the SRT. To decrease the vividness of the scenarios, the second filler task (K-WAI-S-IV–Digit Span Backward) was carried out for approximately 3 min before completing the free recall task. Finally, the past-recollection task was completed. Finally, participants were debriefed and thanked for taking part in the study. The entire experiment lasted for 2 h.

## Result

### General characteristics of groups and test of homogeneity

To investigate between-group differences in demographic variables and pre-test measurements, independent samples $t$-tests and $\chi 2$ tests were performed. None of the pre-test

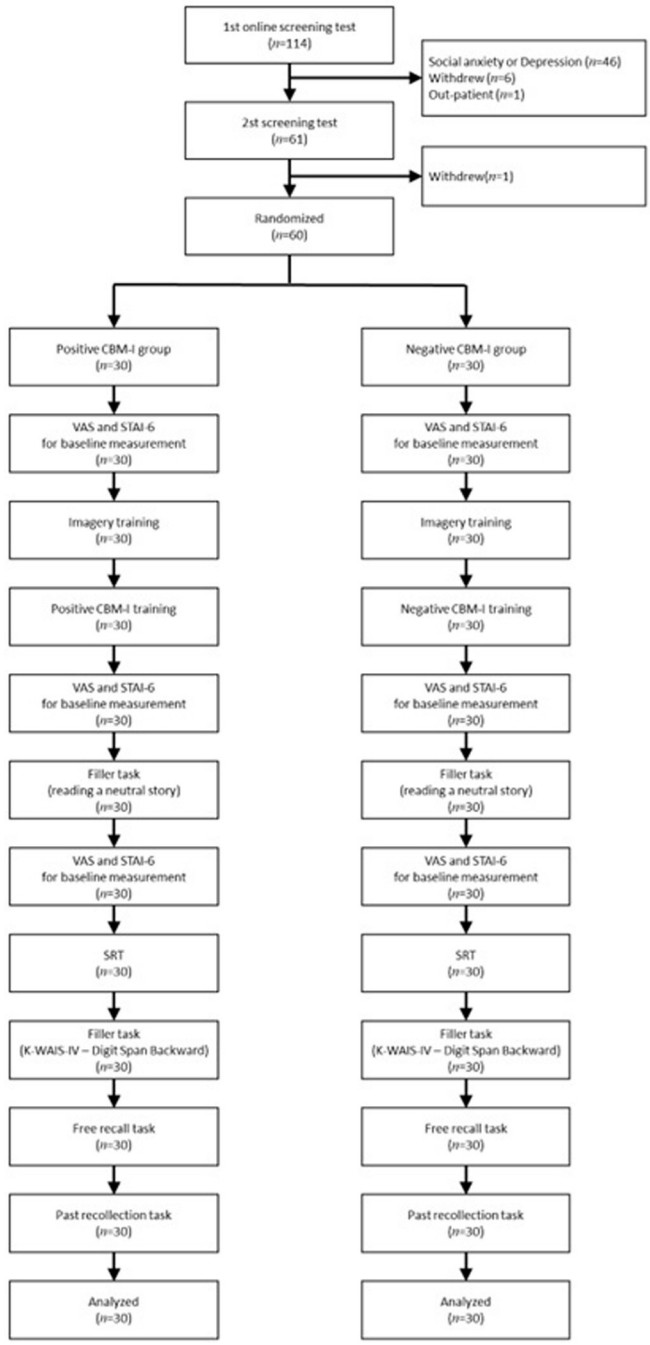

**Fig 1. Flowchart of the experiment for positive and negative CBM-I.**

measurements showed significant differences between the groups: gender, $\chi^2 = .28$, $p = .60$; age, $t = .34$, $p = .73$; happy state mood, $t = -1.78$, $p = .08$; sad state mood, $t = -.64$, $p = .52$.

## The effects of CBM-I on state mood

To examine the effects of CBM-I on participants' mood states, we conducted a 2 (group: positive CBM-I, negative CBM-I) × 2 (time: Pre, Post) × 2 (valence: happiness, sadness) repeated

ANOVA (dependent variables: pre- and post-levels of sadness and happiness). Analyses revealed the three-way interaction of group, time, and valence, $F(1, 56) = 55.52$, $p < .001$, $\eta^2 = .50$. Follow-up analyses inspecting the three-way interactions were conducted separately for happiness and sadness. The results of the follow-up analyses revealed significant differences between the groups in happiness, $t(58) = 2.85$, $p < .01$, Cohen's $d = .73$, and sadness, $t(58) = -4.67$, $p < .001$, Cohen's $d = 1.21$. When paired sample $t$-tests were conducted for each group, happiness increased significantly in the positive group after training ($t(29) = -4.14$, p $< .001$, Cohen's d $= .35$) and decreased significantly in the negative group ($t(29) = 5.69$, p $< .001$, Cohen's d $= .89$). Sadness tended to decrease in the positive group, $t(29) = 2.00$, $p = .055$, Cohen's $d = .29$, and increase significantly in the negative group, $t(28) = -4.41$, $p < .001$, Cohen's $d = .81$.

## The effects of CBM-I on state anxiety

To examine the effects of CBM-I on participants' state anxiety, we conducted a 2 (group: positive CBM-I, negative CBM-I) × 2 (time: Pre, Post) repeated ANOVA (dependent variables: pre-post state anxiety measured with STAI-6). Results revealed that there were the main effect of group, $F(1, 58) = 17.34$, $p < .001$, $\eta^2 = .23$. A two-way interaction of group and time were also significant, $F(1, 58) = 29.87$, $p < .001$, $\eta^2 = .34$. However, the main effect of time was not significant, , $F(1, 58) = 3.12$, $ns$. The follow-up analyses revealed significant differences between the groups; negative state anxiety were higher in the negative CBM-I group than the positive CBM-I group at post-test, $t(58) = -6.26$, $p < .001$, Cohen's $d = 1.62$. When paired sample $t$-test were conducted for each group, the positive group had a lower level of state anxiety at post-test than pre-test, $t(29) = 3.09$, $p < .01$, Cohen's $d = .57$, while the negative group reported increased state anxiety over time, $t(29) = -4.51$, $p < .001$, Cohen's $d = 1.10$.

## The effects of CBM-I on interpretation bias

To investigate the effects of CBM-I on interpretation bias (dependent variables: target positive, target negative, foil positive, foil negative), a generalized linear mixed model (GLMM) was conducted to investigate the interactions between group (positive CBM-I, negative CBM-I), sentence type (target, foil) and valence (positive, negative). Analyses revealed that the three-way interaction between group, valence, and sentence type was not significant, $p = .06$. However, the two-way interaction between group and valence was significant, $p < .001$. Follow-up analyses examining two-way interactions revealed significant group differences for positive, $t(58) = 6.34$, $p < .001$, and negative sentences, $t(58) = -5.13$, $p < .001$. Participants in the positive training group rated higher similarity for positive sentences than for negative sentences (mean estimate = 0.65, Confidence Interval = 0.4504 ~ 0.8663), whereas the negative group did not. To summarize, participants in the positive training group showed positive interpretation bias regardless of sentence type (target and foil), agreeing with the trained valence, whereas participants in the negative training group did not show any effect of training. The results of the mean and standard deviation are provided in Table 1 and, the GLMM analysis are shown in Table 2.

## The effects of CBM-I on memory bias

**Free recall task.** A generalized linear mixed models (GLMMs) were conducted to investigate whether CBM-I training affected the participants' memory bias (see the results of the GLMM in Table 3). Specifically, a two-way interaction between group (positive CBM-I, negative CBM-I) and valence (positive, negative, neutral) was investigated in terms of the total number of intrusions (dependent variables: total number of positive, negative, and neutral

**Table 1. Mean and standard deviation in each group for recognition task, free recall task, and past recollection task.**

| Variable | | Positive CBM-I Group (*n* = 30) | Negative CBM-I Group (*n* = 30) |
|---|---|---|---|
| | | *M* (*SD*) | *M* (*SD*) |
| Recognition task | Target Positive | 3.03 (.47) | 2.37 (.56) |
| | Target Negative | 1.91 (.49) | 2.39 (.65) |
| | Foil Positive | 2.90 (.55) | 2.25 (.50) |
| | Foil Negative | 1.67 (.40) | 2.56 (.59) |
| Free recall task | **Intrusion** | | |
| | positive | 3.53(1.87) | 1.27(1.14) |
| | negative | 2.20(1.22) | 4.00(1.98) |
| | neutral | 4.80(1.85) | 4.77(2.56) |
| | **Content** | | |
| | positive | 4.67(2.60) | .90(1.13) |
| | negative | .83(.95) | 4.50(2.05) |
| | neutral | 4.03(2.09) | 4.13(2.01) |
| | **New idea** | .57(1.76) | .40(.81) |
| | **Error** | .67(.80) | 1.33(1.06) |
| | **Miss** | .47(.82) | .47(.68) |
| Past recollection task | Positive thinking | 9.27(4.22) | 6.47(3.40) |
| | Negative thinking | 2.03(1.61) | 3.30(2.44) |
| | Neutral thinking | 3.27(2.24) | 2.67(2.40) |

intrusion) and content (dependent variables: total number of each positive, negative and neutral content). The result showed a significant two-way interaction between groups and intrusion valence, $p < .001$. Although there was no significant between-group difference in the number of reported neutral intrusions, $t(116) = .07$, $p = .94$, they differed in the number of negative intrusions, $t(116) = -3.79$, $p < .001$, and positive intrusions, $t(116) = 4.77$, $p < .001$. As shown in Fig 2, the positive group reported more positive intrusions than negative group (mean estimate = 2.27, CI = 1.3248 ∼ 3.2085), while the negative group reported more negative intrusions than positive group (mean estimate = -1.80, CI = -2.7419 ∼ -0.8581). Thus, participants in each group reported memory intrusions similar to the valence of their training condition, suggesting that interpretation biases cause memory distortions.

For content, a significant two-way interaction between group and content valence was observed, $p < .001$. No significant between-group difference was found in the number of reported neutral content items, $t(58) = -.19$, $p = .85$, whereas they differed significantly in the number of negative, $t(58) = -8.90$, $p < .001$, and positive content items, $t(58) = 7.27$, $p < .001$. Consistent with the valence of training, the positive group recalled more positive content than the negative group (mean estimate = 3.77, CI = 2.7299 ∼ 4.8034), whereas the negative group recalled more negative content than the positive group (mean estimate = -3.67, CI = -4.4913 ∼ -2.8420).

The groups also differed in the number of errors: the negative group reported more errors than the positive group, $t(58) = -2.75$, $p < .01$. The groups did not differ in the number of new shared $t(58) = .47$, $p = .64$, and misses $t(58) = .00$, $p = 1.00$.

**Past recollection task.** In addition to the memory bias for SRT scenarios, we investigated whether memory bias could be observed in past memories. A generalized linear mixed models (GLMMs) were conducted to investigate whether CBM-I training influenced the participants' past memories. Specifically, a two-way interaction between group (positive CBM-I, negative

**Table 2. GLMM for Effects of CBM-I training on interpretation bias.**

| | Group | Sentence | Valence | Estimate | SE | t-value | P -value |
|---|---|---|---|---|---|---|---|
| Intercept | | | | 2.26 | 0.10 | 23.35 | < .0001 |
| Group | 1 | | | -0.59 | 0.14 | -4.32 | < .0001 |
| Group | 0 | | | 0 | | | |
| Sentence | | 1 | | 0.13 | 0.13 | 1.04 | 0.30 |
| Sentence | | 0 | | 0 | | | |
| Valence | | | 1 | -0.01 | 0.13 | -0.08 | 0.94 |
| Valence | | | 0 | 0 | | | |
| Group*Sentence | 1 | 1 | | 0.11 | 0.18 | 0.64 | 0.53 |
| Group*Sentence | 1 | 0 | | 0 | | | |
| Group*Sentence | 0 | 1 | | 0 | | | |
| Group*Sentence | 0 | 0 | | 0 | | | |
| Sentence*Valence | | 1 | 1 | -0.01 | 0.10 | -0.10 | 0.92 |
| Sentence*Valence | | 1 | 0 | 0 | | | |
| Sentence*Valence | | 0 | 1 | 0 | | | |
| Sentence*Valence | | 0 | 0 | 0 | | | |
| Group*Valence | 1 | | 1 | 1.24 | 0.18 | 6.98 | < .0001 |
| Group*Valence | 1 | | 0 | 0 | | | |
| Group*Valence | 0 | | 1 | 0 | | | |
| Group*Valence | 0 | | 0 | 0 | | | |
| Group*Sentence*Valence | 1 | 1 | 1 | -0.10 | 0.15 | -0.66 | 0.51 |
| Group*Sentence*Valence | 1 | 1 | 0 | 0 | | | |
| Group*Sentence*Valence | 1 | 0 | 1 | 0 | | | |

Group was coded as 1 for positive-CBM -I and 2 for negative CBM-I; The Sentence was coded as 1 for Target sentence and 2 for Foil sentence; Valence was coded as 1 for positive and 2 for negative. SE = Standard Error.

CBM-I) and valence (positive, negative, neutral) was examined in terms of the total number of positive, negative, and neutral memories (dependent variables: positive memories, negative memories, neutral memories) [42]. A gender difference was observed in the total number of positive memories. To account for this difference, gender was included as a covariate in the analysis. Despite controlling for gender as a covariate, the main effect of valence was significant, $p < .001$ and a significant interaction between valence and group was observed, $p < .05$. Follow-up analyses of the interaction revealed significant differences between the groups in the number of positive (mean estimates = 2.80, CI = 0.8199 ∼4.7801) and negative memories (mean difference = -1.27, CI = -2.3338 ∼-0.1995), but no significant difference was found in the number of neutral memories. Specifically, the positive CBM-I group recalled more positive memories, while the negative CBM-I group recalled more negative memories. However, there was no significant difference between the two groups in terms of the number of neutral memories. The results are shown in Table 3.

## Discussion

The main purpose of the present study was to examine whether experimentally induced interpretation bias would affect biases in memory in Korean samples. In doing this, the present study used CBM scenarios that were auditory-specific and with a focus on social anxiety symptoms. The main results showed that interpretation biases can be induced, and induced interpretation biases resulted in training congruent state mood and memory biases on both free-

**Table 3. GLMM for Effects of CBM-I training on free recall and past recollection.**

|  | Group | Valence | Estimates | SE | t-value | P value |
|---|---|---|---|---|---|---|
| **Free recall - Intrusion** |  |  |  |  |  |  |
| Intercept |  |  | 4.77 | 0.34 | 14.18 | < .0001 |
| Group | 1 |  | 0.03 | 0.48 | 0.07 | 0.9444 |
| Group | 0 |  | 0 |  |  |  |
| Valence |  | -1 | -0.77 | 0.45 | -1.70 | 0.0914 |
| Valence |  | 1 | -3.50 | 0.39 | -8.93 | < .0001 |
| Valence |  | 0 | 0 |  |  |  |
| Group*Valence | 1 | -1 | -1.83 | 0.64 | -2.88 | 0.0048 |
| Group*Valence | 1 | 1 | 2.23 | 0.55 | 4.03 | 0.0001 |
| Group*Valence | 1 | 0 | 0 |  |  |  |
| Group*Valence | 0 | -1 | 0 |  |  |  |
| Group*Valence | 0 | 1 | 0 |  |  |  |
| Group*Valence | 0 | 0 | 0 |  |  |  |
| **Free recall-Contents** |  |  |  |  |  |  |
| Intercept |  |  | 4.13 | 0.37 | 11.03 | < .0001 |
| Group | 0 |  | -0.10 | 0.53 | -0.19 | 0.8510 |
| Group | 1 |  | 0 |  |  |  |
| Valence |  | -1 | 0.37 | 0.58 | 0.64 | 0.5264 |
| Valence |  | 1 | -3.23 | 0.67 | -4.80 | < .0001 |
| Valence |  | 0 | 0 |  |  |  |
| Group*Valence | 1 | -1 | -3.57 | 0.81 | -4.38 | < .0001 |
| Group*Valence | 1 | 1 | 3.87 | 0.95 | 4.06 | 0.0002 |
| Group*Valence | 1 | 0 | 0 |  |  |  |
| Group*Valence | 0 | -1 | 0 |  |  |  |
| Group*Valence | 0 | 1 | 0 |  |  |  |
| Group*Valence | 0 | 0 | 0 |  |  |  |
| **Past recollection** |  |  |  |  |  |  |
| Intercept |  |  | 2.67 | 0.42 | 6.29 | < .0001 |
| Group | 1 |  | 0.60 | 0.60 | 1.00 | 0.3210 |
| Group | 0 |  | 0 |  |  |  |
| Valence |  | -1 | 0.63 | 0.63 | 1.01 | 0.3174 |
| Valence |  | 1 | 3.80 | 0.70 | 5.47 | < .0001 |
| Valence |  | 0 | 0 |  |  |  |
| Group*Valence | 1 | -1 | -1.87 | 0.89 | -2.10 | 0.0399 |
| Group*Valence | 1 | 1 | 2.20 | 0.98 | 2.24 | 0.0291 |
| Group*Valence | 1 | 0 | 0 |  |  |  |
| Group*Valence | 0 | -1 | 0 |  |  |  |
| Group*Valence | 0 | 1 | 0 |  |  |  |
| Group*Valence | 0 | 0 | 0 |  |  |  |

Group was coded as 1 for positive-CBM -I and 2 for negative CBM-I; Valence was coded as 1 for positive, 0 for neutral, and -1 for negative. SE = Standard Error.

recall memory and autobiographical memory. These results supported Hirsh and Clark's [19] combined cognitive biases hypothesis that interpretation bias affects memory.

First, the negative group showed a significantly increased negative state mood (sad) compared with the positive group, whereas the positive group showed a significantly increased positive state mood (happy) compared with the negative group. These findings support

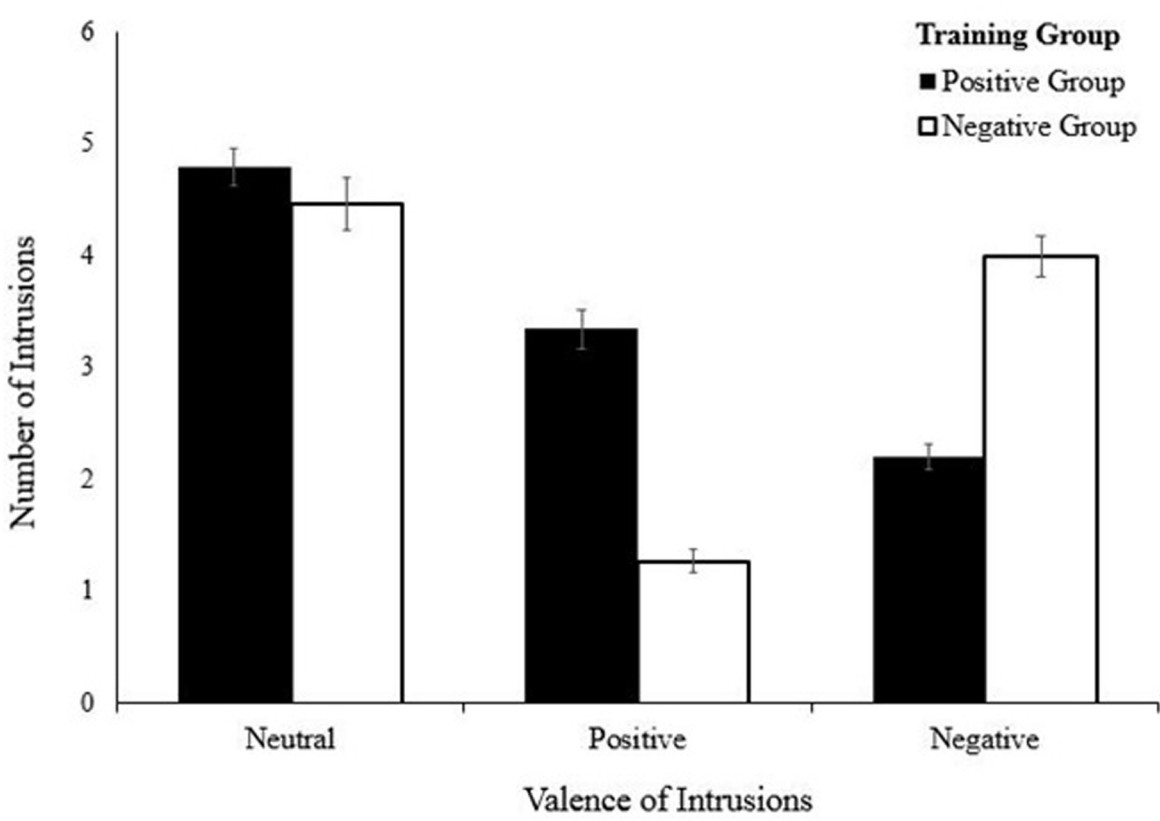

**Fig 2. Mean of neutral, positive, and negative memory intrusion made by each training group during the recall task.** Error bar represent 1 *Standard Error*.

hypothesis 1 and are in line with prior studies that showed that modification training of interpretation bias affects state mood [9,13]. Further, these findings support those in Holmes, Lang [13]: interpretation training using positive mental images in the audio version increases positive affect and reduces anxiety.

Second, the positive group revealed a positive interpretation bias in both the target and foil sentences, whereas the negative group did not show any difference. As such, hypothesis 2 was partially supported. That the positive group displayed positive interpretation bias in both target and foil sentences is consistent with prior studies [9,21]. The cause of the general positive interpretation bias (increased scores in foil sentence) in the positive group can be explained by the information acquisition process [43]. Learning to combine positive interpretations of ambiguous social situation scenarios during the CBM-I training may have shifted to the general interpretations, affecting the high similarity in foil sentences that are less relation to the scenario.

There was no effect of CBM-I training on interpretation bias in the negative group. This result is inconsistent with prior studies that showed that negative CBM-I training can induce negative interpretation bias in a single session [9,21]. The reason for this finding might be that participants in the present study, unlike in prior studies, are healthy people with lower levels of social anxiety and depression. In previous works [9,21], the participants were likely to be recruited without a screening process and so included a mix of healthy people as well as those who tend to be depressed and socially anxious. As the present study selected only healthy participants who have neither social anxiety nor depression through a screening test, they might have had a greater difficulty in having negative mental images. When asked at the debriefing

process whether the training scenarios were commonly experienced in their daily lives and if it was easy to imagine them, most of the participants in negative group reported that the content of the front part of scenarios are similar to their daily life but the negative outcomes at the end of the scenarios are difficult to imagine as it is rather different from their thoughts.

Third, we investigated whether modified interpretation bias could also affect biases in memory. Participants in positive group reported more positive intrusions than the negative group in free recall task, whereas negative group reported more negative intrusions than the positive group, which supports prior results [21,38]. In addition, taking a step beyond prior studies, which only examined the valence of intrusions [21,38], the present study examined the underlying valence of participants' recalled memory contents. The positive group recalled more positive than negative content, similar to intrusions, whereas the negative group recalled more negative than positive content. These results partially support the combined cognitive bias hypothesis argued by Hirsch, Clark [19]. Beyond memory bias related to CBM scenarios, the present study examined whether the CBM-I training affected recall of past autobiographical memories. There were also training-congruent effects of autobiographic memory for the past, with the positive group showed higher positive autobiographical memory whereas the negative group reported higher negative autobiographical memory. These results indicate that induced interpretation bias might affect ongoing memory but also past autobiographic memory.

The limitations of present study and suggestions for further research are as follows. First, the CBM-I trainings in the present study were conducted only with female voices. Among the questions asking about social interaction anxiety [44], one can find "It is difficult to talk with attractive people from the other sex." Such situations are frequent in university students and are one of the leading causes of anxiety [45]. There is a possibility that using female voices would have different effects on the two sexes. Therefore, further research could examine the effects of training more objectively under more controlled conditions where the effects of training are assessed by mixing male and female voices.

Second, we examined the effectiveness of CBM-I training in healthy participants for only one session and could not confirm whether the training effect lasted. Meta-analysis results have shown that training in quasi-clinical groups and increasing the number of sessions can improve the effectiveness of CBM-I [46,47]. Previous studies have demonstrated effectiveness by conducting four sessions of CBM-I training for approximately two weeks [10,48]. In addition, it is necessary to conduct CBM-I training consisting of multiple sessions rather than a single session and to verify the long-term effects through follow-up, given the difficulty in changing core recognition with short-term training. Thus, more studies should be conducted to demonstrate effectiveness by using auditory versions of the mental images-based interpretation bias modification training on quasi-clinical groups and increasing the number of sessions, as well as by including follow-ups. These approaches can increase the effectiveness of CBM-I training and improve the reduction of social anxiety symptoms.

Third, individual differences in memory capacity could not be controlled. Memory capacity may have affected the recall rate of the recall task, and the ability to save the information needed to perform cognitive tasks in a short period of time as well as the functioning capacity to control and manipulate memory may have affected the performance in imagining while listening to scenarios in CBM-I training [49]. A large working memory allows one to handle overall cognitive performance efficiently by enabling one to focus on the stimuli to be addressed and to process control better. Baddeley and Andrade [50] argued that imagination using mental images involves the use of working memory. As the CBM-I method used in the present study involved simultaneously imagining images and listening to sounds, it is possible that listening to and processing sound while still imagining images could have result in

cognitive overload in those with a low working memory. Given the differences between the two groups in the number of errors in the free recall task, it is possible that there were differences in the working memory between the two groups, and that the negative group, which had a higher number of errors, had a smaller working memory. The training might not have showed much effect in the negative group because they were burdened by listening to sounds and recalling unfamiliar negative images. Therefore, it is expected that later studies will be able to demonstrate more clearly the pure effectiveness of CBM-I training by measuring and controlling participants' memory capacities through number memorization tasks in the Wechsler Adult Intelligence Test, the operation span task, or the self-ordered pointing task.

Fourth, to verify the effectiveness of the positive CBM-I training program, we included a comparison group as the negative group, but not a neutral training control group or non-trained control group. So, it is difficult to conclude whether the differences between groups are due to the effects of positive or negative CBM-I training, or the differences in changes caused by both. Therefore, to prove that changes in interpretation and memory bias are inherent effects of positive CBM-I training, the effectiveness of CBM-I training should be verified more clearly by comparing with the results of control groups that receive placebo training, such as neutral interpretation training, or with non-trained groups.

Finally, we acknowledge that the absence of a baseline assessment of interpretation bias is a limitation of our study. Therefore, we urge caution in interpreting the results regarding the effect of CBM-I training on interpretation bias. Despite the random assignment of participants to the positive and negative training groups, we cannot disregard the possibility that there might have been initial differences in positive and negative interpretation tendencies between the groups. To gain a more comprehensive understanding of the impact of CBM-I training on interpretation biases, future research should consider incorporating baseline measurements. This would enable a clearer assessment of the effectiveness of CBM-I interventions in shaping interpretation biases.

Despite these limitations, present study has the following advantages. First, training scenarios were created focusing on social anxiety symptoms. Second, by verifying that a single session CBM-I training affects interpretation, subsequently ongoing and autobiographical memory, the present study provided partial support for the interaction hypothesis proposed by Hirsch and Clark [19] regarding the combined cognitive biases such as memory, interpretation, and attention, interact with one another, thus prolonging mental disorders. Specifically, the study confirmed that a single session of CBM-I training can affect interpretation, ongoing memory, and autobiographical memory. This finding suggests that if the effectiveness of CBM-I training for individuals with social anxiety disorders who exhibit deficit in positive interpretation and memory bias is repeatedly verified, then this technique may be used as therapeutic intervention for social anxiety disorders in the future.

## Supporting information

**S1 Data.**
(XLSX)

## Author Contributions

**Conceptualization:** Jong-Sun Lee.

**Data curation:** Jong-Sun Lee.

**Formal analysis:** Hye Ryeong Park.

**Funding acquisition:** Jong-Sun Lee.

**Investigation:** Hye Ryeong Park, Jong-Sun Lee.

**Methodology:** Hye Ryeong Park, Jong-Sun Lee.

**Project administration:** Jong-Sun Lee.

**Resources:** Jong-Sun Lee.

**Supervision:** Jong-Sun Lee.

**Writing – original draft:** Hye Ryeong Park.

**Writing – review & editing:** Jong-Sun Lee.

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
