## [Decision Letter · Decision Letter 0]

5 Oct 2022

PONE-D-22-04709Induced interpretation bias affects free recall and episodic memory bias in social anxietyPLOS ONE

Dear Dr. Lee,

Thank you for submitting your manuscript to PLOS ONE. After careful consideration, we feel that it has merit but does not fully meet PLOS ONE’s publication criteria as it currently stands. Therefore, we invite you to submit a revised version of the manuscript that addresses the points raised during the review process.

We look forward to receiving your revised manuscript.

Kind regards,

Anindita Bhadra, PhD

Academic Editor

PLOS ONE

Journal Requirements:

“This work was supported by the National Research Foundation of Korea (NRF) grant funded by the Korea government (MSIT) (No. 2018R1C1B5043272).”

“This work was supported by the National Research Foundation of Korea (NRF) grant funded by the Korea government (MSIT) (No. 2018R1C1B5043272). “

“This work was supported by the National Research Foundation of Korea (NRF) grant funded by the Korea government (MSIT) (No. 2018R1C1B5043272).”

6. Please ensure that you include a title page within your main document. You should list all authors and all affiliations as per our author instructions and clearly indicate the corresponding author.

7. Please include a separate caption for each figure in your manuscript.

8. We note you have included a table to which you do not refer in the text of your manuscript. Please ensure that you refer to Table 1 in your text; if accepted, production will need this reference to link the reader to the Table.

Additional Editor Comments (if provided):

This is an interesting piece of work, and I would like to see this published. As you would see, the two expert reviewers have very opposing views on the manuscript. I personally feel that the manuscript needs editing. For example, there are abbreviations being used, without explaining what they are. Some of the references are not in order, as Reviewer 1 has already pointed out. I would also suggest that a flowchart is added to explain the steps of the study. This would provide a good overview to the readers. Further, I would recommend using a generalized linear model, instead of the mixed-model ANOVA for the analysis.

Reviewers' comments:

Reviewer's Responses to Questions

**Comments to the Author**

1. Is the manuscript technically sound, and do the data support the conclusions?

Reviewer #1: Partly

Reviewer #2: Yes

2. Has the statistical analysis been performed appropriately and rigorously? 

Reviewer #1: No

Reviewer #2: Yes

3. Have the authors made all data underlying the findings in their manuscript fully available?

Reviewer #1: No

Reviewer #2: Yes

4. Is the manuscript presented in an intelligible fashion and written in standard English?

Reviewer #1: No

Reviewer #2: Yes

5. Review Comments to the Author

Reviewer #1: Overall, the manuscript needs significant improvement in terms of clarity and writing coherence. Also, there are some grammatical and writing errors that getting help from a native English speaker who is familiar with psychological research to proofread the work would help with that.

Abstract

1. Please mention the number of participants in the abstract.

Intro

1. I suggest that authors summarize/cut some of the social anxiety disorder prevalence-related information in the first paragraph of the introduction to make this part more to the point.

2. Please check the in-text citations as currently some of them seems incorrect, for example, Amir, Foa (7).

3. In line 30, in describing the Amir et al (7) study, the authors didn’t clarify whether the social anxiety groups in this study were clinical participants or subclinical individuals with elevated social anxiety. Please specify for this study and throughout the manuscript what is meant by “group with social anxiety”.

4. The definition of the CBM-I in lines 41-43 is not clear, please provide a more informative definition.

5. It is unclear what is meant by “distortion of memory” in line 76. Please specify.

6. In line 79, it is unclear what the pronoun “them” is referring to.

7. It is better to avoid using the word “prove” for psychological hypotheses.

8. The paragraph that sums up the rationale for the study is not clear, and needs to be re-written. It seems somewhat disconnected from previous paragraphs, and some parts of it are difficult to follow and understand. For example, I suggest that authors describe the auditory method of reading the scenarios before this paragraph. Because this is an important point in the design of the current study and the first time it appears about the end of the introduction without it being sufficiently described before.

Method

1. The first time you use an abbreviation, spell out the full term and put the abbreviation in parentheses.

2. Please provide more information for each questionnaire, what the questionnaire was designed to measure, if it has been reported as a valid and reliable measure by previous researchers, and if the psychometric properties of the Korean version of the questionnaire have been examined.

3. The similarity rating task description is not clear enough. It is not clear what the authors meant by the "title of scenarios". Moreover, the order of describing different parts of the task is not optimal, for example, the authors first wrote that the participants were asked to complete the incomplete word in the last phrase of the scenario, and then the scenarios. Generally, this section needs to be re-written more clearly with providing an example trial/scenario to help the readers to follow and understand the task.

4. What was the neutral interpretation question in the SRT task? Please provide more information on that.

5. In the recognition phase of the SRT how many trials/scenarios were presented for recognition? Was it the case that each scenario's title was presented with all four types of sentences? Please clarify.

6. Please explain the indices that were computed from the SRT task in the same section you described the task.

7. In the free recall task description, the authors named "the new content that did not apply to the scenarios or recognition task sentences" as intrusions! I don't think we can call these incorrect memories as intrusions. Intrusions are involuntarily thoughts and images that are irrelevant and interrupting to the ongoing activity, while in the recall task participants themselves were actively trying to remember content related to the scenarios, therefore even if they remembered something incorrect these could not be named intrusions.

8. The difference between what is defined as “intrusions” and "new ideas" is not clear, and it seems these two categories overlap largely, which is a methodological issue.

Results

1. The authors only analyzed the second part (recognition phase) of the SRT task, while the first part that asked participants to complete the last word of the ambiguous neutral sentences has not been analyzed. It is possible that the CBM-I groups significantly differed in interpreting the sentences of the first phase of the SRT task.

2. Please specify the dependent variable for each ANOVA analysis.

3. For the free recall task, it is unclear why the authors based their main analyses on intrusion that they themselves defined as "reporting of new content that did not apply to the scenarios or recognition task sentences". What were the reasons that the authors did not use the degree to which participants of each group remembered accurate negative or positive material?

4. For the second ANOVA in the Free recall task analysis, it is unclear what exactly meant by "content". Was it everything that participants remembered or a specific part of what they remembered?

5. Why the authors controlled for the role of gender in the ANOVA analysis for past thinking task, while the same had not been done for previous analyses?

Discussion

1. The authors wrote that “a part of the combined cognitive biases hypothesis” was confirmed. What part of this hypothesis they exactly referring to? I think they meant the combined cognitive biases hypothesis for interpenetration bias and memory bias, so if this is the case, specify that instead of saying "a part" that is vague and unclear!

2. Line 401, control group are described as "healthy people without any social anxiety and depression". The sentence seems not true given the dimensional nature of social anxiety and depression.

Reviewer #2: Well written and without error. I would ask the authors to make one more check that all word spacing is correct, all terms are properly expanded (particularly DSM-5 on page 1, sentence 1) and the use of references in the paper are correct as per the journal requirements. I use the ACS style guide more than any other reference and yours may be perfectly correct, but simply different than what I am familiar with.

6. PLOS authors have the option to publish the peer review history of their article (what does this mean?). If published, this will include your full peer review and any attached files.

Reviewer #1: No

Reviewer #2: No

---

## [Author Response · Author response to Decision Letter 0]

15 Mar 2023

Response to Peer Review Comments.

Dear Professor Anindita Bhadra, 

Thank you for your editorial letter regarding the manuscript entitled “Induced interpretation bias affects free recall and episodic memory bias in social anxiety”. We are very pleased with another opportunity for revising our manuscript. We have tried to do our best to address the concerns each reviewer raised. We have also revised our manuscript as per your advice below. Especially, we have updated statistical analyses using the generalized linear mixed models (GLMM) as you advised. I sincerely apologize for the delay in submitting the revised manuscript. I want to assure you that I took the necessary time to learn the statistical method you suggested, and to thoroughly re-analyze and derive the results to ensure their accuracy and validity. Thank you for your patience and understanding."

Journal Requirements:

Response. we have revised the manuscript as you requested. 

Response. we have updated the method section to include detailed information about participant consent. 

On the day of the experiment, the participants completed the written consent form followed by the secondary screening survey.

“This work was supported by the National Research Foundation of Korea (NRF) grant funded by the Korea government (MSIT) (No. 2018R1C1B5043272).”

Response. as per your suggestion, we have changes to the Funding statement, whihch is now updated as follows: 

“This work was supported by the National Research Foundation of Korea (NRF) grant funded by the Korea government (MSIT) (No. 2018R1C1B5043272). There was no additional external or internal funding received for this study”.

“This work was supported by the National Research Foundation of Korea (NRF) grant funded by the Korea government (MSIT) (No. 2018R1C1B5043272). “

“This work was supported by the National Research Foundation of Korea (NRF) grant funded by the Korea government (MSIT) (No. 2018R1C1B5043272).”

Response. we have removed the Funding statement, including any acknowledgements of funding sources, from the manuscript. Please update the Funding statements below accordingly.

“This work was supported by the National Research Foundation of Korea (NRF) grant funded by the Korea government (MSIT) (No. 2018R1C1B5043272). There was no additional external or internal funding received for this study”.

Response. we have uploaded the dataset as Supporting information. 

6. Please ensure that you include a title page within your main document. You should list all authors and all affiliations as per our author instructions and clearly indicate the corresponding author.

Response. we have completed it as per your instructions

7. Please include a separate caption for each figure in your manuscript.

Response. we have included separate captions for each of the figures, namely, figure 1 and figure 2. 

8. We note you have included a table to which you do not refer in the text of your manuscript. Please ensure that you refer to Table 1 in your text; if accepted, production will need this reference to link the reader to the Table.

Response. we have included descriptions of Table 1, Table 2, and Table 3 in the manuscript. 

Additional Editor Comments (if provided):

This is an interesting piece of work, and I would like to see this published. As you would see, the two expert reviewers have very opposing views on the manuscript. I personally feel that the manuscript needs editing. For example, there are abbreviations being used, without explaining what they are. Some of the references are not in order, as Reviewer 1 has already pointed out. I would also suggest that a flowchart is added to explain the steps of the study. This would provide a good overview to the readers. Further, I would recommend using a generalized linear model, instead of the mixed-model ANOVA for the analysis.

Response: "In response to your helpful comments, we have made two significant changes to the manuscript. Firstly, we have included a flowchart as Figure 1. Secondly, we have used a GLMM in place of a mixed model ANOVA to analyze the SRT and free recall and past episodic memory bias data. Thank you for your valuable feedback."

Comments to the Author

Reviewer #1: Overall, the manuscript needs significant improvement in terms of clarity and writing coherence. Also, there are some grammatical and writing errors that getting help from a native English speaker who is familiar with psychological research to proofread the work would help with that.

Abstract

1. Please mention the number of participants in the abstract.

Response. As per your comments, we have updated the number of participants in the abstract. Thank you for your valuable feedback. 

The combined effect of each cognitive bias is known to play a causal role in the etiology and maintenance of social anxiety. However, little is known about how each type of bias acts during social anxiety. The present study aimed to investigate whether experimentally induced interpretation bias using the cognitive bias modification (CBM) paradigm would influence free recall and episodic memory biases in a Korean sample. A total of 61 participants were randomly assigned to either a positive (n = 30) or negative (n = 31) CBM group.

Intro

1. I suggest that authors summarize/cut some of the social anxiety disorder prevalence-related information in the first paragraph of the introduction to make this part more to the point.

Response. Thank you for your helpful feedback. We have summarized the first paragraph related to social anxiety disorder. 

2. Please check the in-text citations as currently some of them seems incorrect, for example, Amir, Foa (7).

Response. We have revised the in-text citations throughout the manuscript in consultation with the editorial office. 

3. In line 30, in describing the Amir et al (7) study, the authors didn’t clarify whether the social anxiety groups in this study were clinical participants or subclinical individuals with elevated social anxiety. Please specify for this study and throughout the manuscript what is meant by “group with social anxiety”.

Response. Thank you for your comments. We have clarified the characteristics of the participants as follows:

For example, Amir, Foa and Coles [6] presented ambiguous social or non-social situations to three groups; clinical groups of people with generalized social phobia and obsessive-compulsive disorder, as well as a control group without social anxiety. 

4. The definition of the CBM-I in lines 41-43 is not clear, please provide a more informative definition.

Response. Thank you for your feedback. We have provided a clearer description of CBM-I.

Cognitive Bias Modification of interpretation (CBM-I) aims to positively correct negative biases in individuals with anxiety or depression, by providing repetitive computer-based tasks for interpreting ambiguous social situations in a more flexible and positive way [9]. Participants can complete the training task at home or wherever they prefer [10].

5. It is unclear what is meant by “distortion of memory” in line 76. Please specify.

Response. We added detailed information that reflects the result of the experiment. 

On recalling the scenarios, the social anxiety groups tended to report a higher percentage of false errors with emotionally negative content compared with the control group.

6. In line 79, it is unclear what the pronoun “them” is referring to.

Response. We revised ‘them’ into ‘ambiguous social scenarios. 

One group of participants was trained to interpret ambiguous social scenarios positively, and the other, negatively.

7. It is better to avoid using the word “prove” for psychological hypotheses.

Response. We have updated the word in our manuscript from “prove” to “supported” for all hypotheses throughout the manuscript. For instance, 

These results partially supported Hirsch and Clark’s [1] combined cognitive biases hypothesis that interpretation bias affects memory.

8. The paragraph that sums up the rationale for the study is not clear, and needs to be re-written. It seems somewhat disconnected from previous paragraphs, and some parts of it are difficult to follow and understand. For example, I suggest that authors describe the auditory method of reading the scenarios before this paragraph. Because this is an important point in the design of the current study and the first time it appears about the end of the introduction without it being sufficiently described before.

Response. Thank you for your feedback. We have added a more detailed explanation of CBM-I, including the use of auditory imagery in the intervention, to the manuscript. 

Cognitive Bias Modification of interpretation (CBM-I) aims to positively correct negative biases individuals with anxiety or depression, by providing repetitive computer-based tasks for interpreting ambiguous social situations in a more flexible and positive way [9]. Participants can complete the training task at home or wherever they prefer. Mathews and Mackintosh [10]found that participants in the positive CBM-I group showed significantly reduced levels of anxiety as well as increased positive interpretation bias compared with participants in the negative CBM-I group, confirming the causal relationships between cognitive bias and mood. Mathews and Mackintosh [10] argued that CBM-I, which induces positive interpretations in social anxiety, may have therapeutic implications that reduce anxiety caused by negative interpretation bias. Indeed, many prior studies with the subclinical population with social anxiety symptoms have since demonstrated iterative CBM-I training can modify interpretation bias for social situations and then affect the change of symptoms [9, 11, 12]. Previous studies have explored the optimal mode of delivering CBM-I intervention that involve short stories based on daily life experiences. It has been hypothesized that imagining scenarios mentally is more effective in influencing mood compared to verbal reasoning. Indeed, several studies have demonstrated that auditory CBM-I that utilizes imagery has a greater impact on mood and interpretation biases compared to visually presented CBM-I intervention [13, 14] [15]. 

Method

1. The first time you use an abbreviation, spell out the full term and put the abbreviation in parentheses.

Response. We have thoroughly checked and spelt out the full name. 

A total of 114 undergraduate and graduate students completed an online screening survey using the Center for Epidemiological Studies-Depression (CES-D), Social Avoidance Distress Scale (SADS), and Brief version of Fear of Negative Evaluation Scale (B-FNE).

2. Please provide more information for each questionnaire, what the questionnaire was designed to measure, if it has been reported as a valid and reliable measure by previous researchers, and if the psychometric properties of the Korean version of the questionnaire have been examined.

Response. In response to your comments, we have included additional information about he the purpose of each questionnaire, its previous use as valid and reliable measure, and whether its psychometric characteristics were evaluated in the Korean version of the questionnaire. 

Questionnaires

Center for Epidemiological Studies-Depression (CES-D). 

The CES-D [23, 24] was utilized during the screening phase to measure the degree of depression experienced in the previous week, using a four-point Likert scale (0 = extremely rare to 3 = most likely). It has demonstrated strong psychometric properties, including a high internal consistency coefficient (Radloff [23] Cronbach’s α = .85; Cho and Kim [24] Cronbach’s α = .91) and satisfactory test-retest reliability (Radloff [23] r = .57; Cho and Kim [24] r = .68). Our study yielded a Cronbach's α of .70.

Social Avoidance and Distress Scale (SADS).

The SADS [25, 26] was used in the screening phase to evaluate the anxiety and inclination of participants to avoid social situations on a five-point Likert scale (ranging from 1 = do not agree at all to 5 = strongly agree). It has good psychometric properties, including a high internal consistency coefficient (Lee and Choi [26] Cronbach’s α = .92) and satisfactory test-retest reliability (Watson and Friend [25] r = .68; Lee and Choi [26] r = .88). Our Cronbach's α was .88.

Brief Version of Fear of Negative Evaluation Scale (B-FNE). 

The B-FNE [25-27] was used during the screening phase to assess the fear of being negatively evaluated by others, which is a key characteristic of social anxiety. Response were given on a five-point Likert scale ranging from (not at all) to 5 (very much). The questionnaire has been reported to have good psychometric properties, including high internal consistency coefficient (Lee and Choi [26]; Leary Lee and Choi (26), [27] Cronbach’s α = .90) and good test-retest reliability (Watson and Friend [25] r = .78; Lee and Choi [26] r = .80; Leary [27] r = .75). Our study obtained a Cronbach’s α of .89. 

Visual Analogue Scale (VAS). 

To assess participants’ state mood, they were asked to rate their mood before and after the training and the first filler task. The VAS was used as a self-report measure of current emotions and pain. The VAS is a simple tool that allows individuals to rate their subjective state [28] by marking an 'X' on a horizontal bar ranging from 0 (not at all) to 10 (extremely) for negative (sadness) and positive (happiness) emotions. This scale is ideal for self-reporting as the response burden is low [29]. 

Six-item short version of the state-trait anxiety inventory (STAI-6). 

STAI-6 [30] is the shorter version of the State-Trait Anxiety Inventory-State (STAI-S) [31]. It assesses the current level of anxiety using a four-point Likert scale (1 = not at all to 4 = strongly agree). Compared to STAI-S, it is a concise, reliable, and valid measure that is sensitive to changes in anxiety [30, 32]. Previous studies have reported high internal consistency coefficient (Tluczek, Henriques [32] Cronbach’s α ranging from .74 to .82). In our study, the Cronbach’s α was .74 before training, .80 after training, and .77 after the filler tasks.

3. The similarity rating task description is not clear enough. It is not clear what the authors meant by the "title of scenarios". Moreover, the order of describing different parts of the task is not optimal, for example, the authors first wrote that the participants were asked to complete the incomplete word in the last phrase of the scenario, and then the scenarios. Generally, this section needs to be re-written more clearly with providing an example trial/scenario to help the readers to follow and understand the task.

Response. We have revised and rewritten the description of the SRT with an example. 

4. What was the neutral interpretation question in the SRT task? Please provide more information on that.

Response. We have provided the rewritten description of the SRT with an example. 

5. In the recognition phase of the SRT how many trials/scenarios were presented for recognition? Was it the case that each scenario's title was presented with all four types of sentences? Please clarify.

Response. We have provided a more detailed description of the SRT. This includes an example to help illustrate how the task is performed by participants. 

6. Please explain the indices that were computed from the SRT task in the same section you described the task.

Response. The indices used in the final analysis have been included at the end of the SRT description. 

Similarity Rating Task (SRT). 

The SRT was used in this study to assess the participants’ interpretation bias after CBM training. The procedure and format of the SRT were based on the previous studies [35, 36]. The SRT comprises two phases: encoding and recognition. In the encoding phase, below the title of each story, participants were presented with 10 scenarios, each with an ambiguous and neutral descriptions ending in a word fragment completion, followed by a neutral comprehension question (‘yes or no’ answer) regarding the content of each description. An example is as follows. 

<Presentation of liberal arts class>

You give a presentation in a liberal arts class.

After the presentation, some people ask you questions.

As you answer the last question, the classroom goes sil__nt.

Did you answer the question?

[yes or no]

In the following recognition phase, each scenario’s title was presented along with four types of sentences in random order: (1) positive sentence related to the scenario (target positive; i.e., people are happy with my answer), (2) negative sentence related to the scenario (target negative; i.e., people are so frustrated with my answer), (3) positive sentence unrelated to the scenario (foil positive; i.e., you finish your presentation on time.), and (4) negative sentence unrelated to the scenario (foil negative; i.e., you don't finish your presentation on time). Participants were asked to rate how similar each sentence was to the scenario they had read in the encoding phase on a four-point Likert scale (1 = very different; 2 = fairly different; 3 = fairly similar; 4 = very similar). The target sentences were either positive or negative interpretation related to the social situation scenario, the foil sentences were generally positive or negative [37]. The four types of sentences were fixed for each scenario and were presented in a random order. This study included 40 trials, with each scenario having four different trials consisting of a target positive, a target negative, a foil positive, and a foil negative interpretation. 

The SRT was conducted using the E-prime 2.0 software and displayed on a 16-inch laptop screen with a white background and navy-colored Arial font. Before actual test, participants completed five practice trials. The mean score for each type of the sentence (target positive, target negative, foil positive, and foil negative) was calculated and used in the final analysis. 

7. In the free recall task description, the authors named "the new content that did not apply to the scenarios or recognition task sentences" as intrusions! I don't think we can call these incorrect memories as intrusions. Intrusions are involuntarily thoughts and images that are irrelevant and interrupting to the ongoing activity, while in the recall task participants themselves were actively trying to remember content related to the scenarios, therefore even if they remembered something incorrect these could not be named intrusions.

Response. We appreciate your feedback. The scoring procedure for intrusions was informed by previous research conducted by Salemink, Hertel, and Makintosh (2010), as well as Joormann, Waugh, and Gotlib (2015). Specifically, Salemink, Hertel and Mackintosh (2010) defined intrusions as “terms that had not been presented originally”, and we adopted this definition after consulting with Joormann prior to the study. During the SRT recall task (i.e., both encoding and recognition phases), participants occasionally reported irrelevant and distracting information. Whether these intrusions were intentional or unintentional remains unclear. However, we scored intrusions according to the definition established in previous research, namely as irrelevant content that was not related to the SRT descriptions.

8. The difference between what is defined as “intrusions” and "new ideas" is not clear, and it seems these two categories overlap largely, which is a methodological issue.

Response. Thank you for the clarification. Based on the scoring system used by Salemink, Hertel, and Mackintosh (2010) and Joormann, Waugh, and Gotlib (2015), intrusions were defined as any new terms or ideas that were not originally presented in the SRT task. On the other hand, new autobiographical memory ideas were categorized separately from intrusions, namely new ideas.

Results

1. The authors only analyzed the second part (recognition phase) of the SRT task, while the first part that asked participants to complete the last word of the ambiguous neutral sentences has not been analyzed. It is possible that the CBM-I groups significantly differed in interpreting the sentences of the first phase of the SRT task.

Response. Thank you for the suggestion. In the first part of the SRT, participants were presented with a script that included an empty word, which they were required to fill in (e.g., "As you answer the last question, the classroom goes sil__nt"), followed by a question to confirm their understanding of the script (e.g., "Did you answer the question? Yes or No?"). We analyzed the responses of the first part of the SRT and found no significant differences between the two groups. The correct response rate was almost identical for both groups.

2. Please specify the dependent variable for each ANOVA analysis.

Response. Thank you for the feedback. We have revised the manuscript and added a specification of the dependent variable for each ANOVA analysis.

The effects of CBM-I on state mood 

To examine the effects of CBM-I on participants’ mood states, we conducted a 2 (group: positive CBM-I, negative CBM-I) × 2 (time: Pre, Post) × 2 (valence: happiness, sadness) repeated ANOVA (dependent variables: pre- and post-levels of sadness and happiness).

The effects of CBM-I on interpretation bias

To investigate the effects of CBM-I on interpretation bias, a 2 (group: positive CBM-I, negative CBM-I) × 2 (sentence type: target, foil) × 2 (valence: positive, negative) mixed-model ANOVA was conducted on SRT (dependent variables: target positive, target negative, foil positive, foil negative).

Free recall task

The aim of the present study was to investigate whether CBM-I training affected the participants’ memory bias. We conducted a 2 (group: positive CBM-I, negative CBM-I) × 3 (valence: positive, negative, neutral) mixed model ANOVA each for the total number of intrusions and content (dependent variables related to intrusion included each positive, negative and neutral intrusion, while the dependent variables related to content included each positive, negative and neutral content).

Past thinking task

In addition to the memory bias for SRT scenarios, we investigated whether memory bias could be observed in past memories. Specifically, a 2 (group: positive CBM-I, negative CBM-I) × 3 (valence: positive, negative, neutral) mixed-model ANOVA was performed for the total number of positive, negative, and neutral memories [41] (dependent variables: the total number of each positive. negative memories, neutral episodic memories).

3. For the free recall task, it is unclear why the authors based their main analyses on intrusion that they themselves defined as "reporting of new content that did not apply to the scenarios or recognition task sentences". What were the reasons that the authors did not use the degree to which participants of each group remembered accurate negative or positive material?

Response. Thank you for your comments. The scoring procedure for intrusions was informed by previous research conducted by Salemink, Hertel, and Makintosh (2010), as well as Joormann, Waugh, and Gotlib (2015). Specifically, Salemink, Hertel and Mackintosh (2010) defined intrusions as “terms that had not been presented originally”, and we adopted this definition after consulting with Joormann prior to the study. During the SRT recall task (i.e., both encoding and recognition phases), participants occasionally reported irrelevant and distracting information. Whether these intrusions were intentional or unintentional remains unclear. However, we scored intrusions according to the definition established in previous research, namely as irrelevant content that was not related to the SRT descriptions.

We did not come across any explicit criteria defining the accuracy of positive or negative material recalled by the participants. Rather, as we addressed in our previous responses, we classified the emotional valence of all the recalled content as positive, negative, or neutral. As an illustration, a participant's statement such as "I was nervous and did my presentation awkwardly, but I managed to complete it well and received good feedback" was regarded as positive, despite containing a negative sentence. However, we did not evaluate the accuracy of the participants' responses.

4. For the second ANOVA in the Free recall task analysis, it is unclear what exactly meant by "content". Was it everything that participants remembered or a specific part of what they remembered?

Response. Thank you for your comments. In the free recall task, as mentioned in the manuscript, we coded the valence of all the content recalled by participants as positive, negative, or neutral. For example, a statement such as “I was nervous and did my presentation awkwardly, but I managed to completed it well and received good feedback” which included a negative sentence but had overall positive content, was coded as positive.

5. Why the authors controlled for the role of gender in the ANOVA analysis for past thinking task, while the same had not been done for previous analyses?

Response. Thank you for your feedback. As there was a significant difference in gender for past thinking task, but not for other tasks, we conducted a follow-up analysis by controlling for gender. However, the results of the follow-up analysis remained unchanged. 

Discussion

1. The authors wrote that “a part of the combined cognitive biases hypothesis” was confirmed. What part of this hypothesis they exactly referring to? I think they meant the combined cognitive biases hypothesis for interpenetration bias and memory bias, so if this is the case, specify that instead of saying "a part" that is vague and unclear!

Response. Thank you for feedback. We have now provided specific information on which component of the combined cognitive bias is supported by our findings.

The main purpose of the present study was to examine whether experimentally induced interpretation bias would affect biases in memory in Korean samples. In doing this, the present study used CBM scenarios that were auditory-specific and with a focus on social anxiety symptoms. The main results showed that interpretation biases can be induced, and induced interpretation biases resulted in training congruent state mood and memory biases on both free-recall memory and autobiographical memory. These results supported Hirsh and Clark’s [1] combined cognitive biases hypothesis that interpretation bias affects memory.

2. Line 401, control group are described as "healthy people without any social anxiety and depression". The sentence seems not true given the dimensional nature of social anxiety and depression.

Response. Thank you for the feedback. We have revised the original phrase ‘without any social anxiety and depression’ to ‘with lower levels of social anxiety and depression’.

The reason for this finding might be that participants in the present study, unlike in prior studies, are healthy people with lower levels of social anxiety and depression. In previous works (10, 18), the participants were likely to be recruited without a screening process and so included a mix of healthy people as well as those who tend to be depressed and socially anxious.

Reviewer #2: Well written and without error. I would ask the authors to make one more check that all word spacing is correct, all terms are properly expanded (particularly DSM-5 on page 1, sentence 1) and the use of references in the paper are correct as per the journal requirements. I use the ACS style guide more than any other reference and yours may be perfectly correct, but simply different than what I am familiar with.

Response. Thank you for your feedback. We have made the necessary changes to the reference citation to adhere to the Plos style guidelines.

---

## [Decision Letter · Decision Letter 1]

22 May 2023

PONE-D-22-04709R1

Induced interpretation bias affects free recall and episodic memory bias in social anxiety

PLOS ONE

Dear Dr. Lee,

Thank you for submitting your manuscript to PLOS ONE. After careful consideration, we feel that it has merit but does not fully meet PLOS ONE’s publication criteria as it currently stands. Therefore, we invite you to submit a revised version of the manuscript that addresses the points raised during the review process.

We look forward to receiving your revised manuscript.

Kind regards,

Anindita Bhadra, PhD

Academic Editor

PLOS ONE

Additional Editor Comments:

As you would see, the Reviewer 1 has one major concern that needs to be addressed. I am happy that the reviewer finds the revised version of the manuscript much improved. I would request you to address this at your earliest and submit a revised version of your manuscript.

Reviewers' comments:

Reviewer's Responses to Questions

**Comments to the Author**

1. If the authors have adequately addressed your comments raised in a previous round of review and you feel that this manuscript is now acceptable for publication, you may indicate that here to bypass the “Comments to the Author” section, enter your conflict of interest statement in the “Confidential to Editor” section, and submit your "Accept" recommendation.

Reviewer #1: (No Response)

2. Is the manuscript technically sound, and do the data support the conclusions?

Reviewer #1: Yes

3. Has the statistical analysis been performed appropriately and rigorously? 

Reviewer #1: Yes

4. Have the authors made all data underlying the findings in their manuscript fully available?

Reviewer #1: Yes

5. Is the manuscript presented in an intelligible fashion and written in standard English?

Reviewer #1: Yes

6. Review Comments to the Author

Reviewer #1: The authors have done an excellent job of addressing the comments and significantly improving the new version of the manuscript. Congratulations to the authors for conducting this interesting study. However, there is one major issue and two minor issues that I would like to raise regarding the new version of the work. I believe that after addressing the remaining issues, the manuscript will be ready for publication.

Major issue

There is a misinterpretation of the results related to the CBM-I training effects. In the section “The effects of CBM-I on interpretation bias” the authors report: "Participants in the positive training group rated higher similarity for positive sentences than for negative sentences (mean estimate = 0.65, Confidence Interval = 0.4504∼0.8663), whereas the negative group did not." The authors then summarize and conclude that “To summarize, participants in the positive training group showed positive interpretation bias regardless of sentence type (target and foil), agreeing with the trained valence, whereas participants in the negative training group did not show any effect of training”.

This is a problematic interpretation of the finding. There was no assessment of baseline interpretation biases before the CBM-I training in the current study, which makes it impossible to conclude that the negative CBM-I was not successful in influencing Interpretation Biases (IB) or that the positive CBM-I was successful in changing participants’ IB. This might be the case that all participants had a significantly more positive than negative IB at the baseline, especially that the researchers recruited participants with low levels of social anxiety for their study. If that is true (which we don’t know due to a lack of baseline assessment of IB), the negative CBM-I was indeed effective in reducing participants’ positive interpretation biases to a level that no significant difference was found between positive and negative interpretation biases in the negative group. Also, it would be the case that because participants already had more positive than negative IB, the positive CBM-I either did not impact their IBs or slightly increased them. Therefore, this cannot be concluded that the positive or negative CBM-I trainings were successful or not in changing participants’ interpretation biases as we don’t know participants’ initial baseline IB patterns. Although given that participants were randomly assigned to the CBM-I groups, it is very likely that their baseline IB were similar, and it can be concluded that their significant difference after the CBM-I training is due to the training, but still we don’t know if both groups had more positive than negative IB before the CBM-I training (in that case negative CBM-I was effective), or more negative than positive IB (in that case positive CBM-I was effective), or equal positive and negative IB.

Likewise, if authors hypothesize and expect that participants in the negative CBM-I show more negative than positive IB and those in the positive CBM-I show more positive than negative IB, they can say that these findings did not fully supported their hypothesis, but it is still incorrect to conclude that positive CBM-I was successful in changing IB and negative CBM-I failed to change participants IB for the same reason that I explained earlier.

The same problem can be seen in the discussion section too where the authors wrote:

“There was no effect of CBM-I training on interpretation bias in the negative group….”

They also write: “Second, the positive group revealed a positive interpretation bias in both the target and foil sentences, whereas the negative group did not show any difference before and after CBM training”. It is important to note that in this sentence, they mention “before and after CBM training,” but the interpretation bias assessment occurred only after the CBM-I training in the study.

I think this problem in the interpretation of the effects of the CBM-I training is very important and needs to be addressed throughout the manuscript.

Minor issues

1- Please specify the name of the biases in the first two sentences of the abstract i.e. interpretation and memory. Currently, the way the abstract starts talks about "each bias" and "each type of bias" which seems not clear and can include any type of different bias.

2- The order of two sentences in the introduction needs to be changed. Specifically, the sentence that starts with "Mathews and Mackintosh [10] found that participants in the positive CBM-I group...." seems to belong after its next sentence i.e. "Mathews and Mackintosh [10] argued that CBM-I, which induces positive interpretations in social anxiety, may have ..." not before it.

7. PLOS authors have the option to publish the peer review history of their article (what does this mean?). If published, this will include your full peer review and any attached files.

Reviewer #1: **Yes: **Mahdi Mazidi

---

## [Author Response · Author response to Decision Letter 1]

26 May 2023

Response to Peer Review Comments.

Dear Professor Anindita Bhadra, 

Thank you for your editorial letter regarding the manuscript entitled “Induced interpretation bias affects free recall and episodic memory bias in social anxiety”. We are very pleased with another opportunity for revising our manuscript. We have tried to do our best to address the concerns each reviewer raised. We have also revised our manuscript as per your advice below. Thank you for your patience and understanding."

Reviewer #1: The authors have done an excellent job of addressing the comments and significantly improving the new version of the manuscript. Congratulations to the authors for conducting this interesting study. However, there is one major issue and two minor issues that I would like to raise regarding the new version of the work. I believe that after addressing the remaining issues, the manuscript will be ready for publication.

Major issue:

There is a misinterpretation of the results related to the CBM-I training effects. In the section “The effects of CBM-I on interpretation bias” the authors report: "Participants in the positive training group rated higher similarity for positive sentences than for negative sentences (mean estimate = 0.65, Confidence Interval = 0.4504∼0.8663), whereas the negative group did not." The authors then summarize and conclude that “To summarize, participants in the positive training group showed positive interpretation bias regardless of sentence type (target and foil), agreeing with the trained valence, whereas participants in the negative training group did not show any effect of training”.

This is a problematic interpretation of the finding. There was no assessment of baseline interpretation biases before the CBM-I training in the current study, which makes it impossible to conclude that the negative CBM-I was not successful in influencing Interpretation Biases (IB) or that the positive CBM-I was successful in changing participants’ IB. This might be the case that all participants had a significantly more positive than negative IB at the baseline, especially that the researchers recruited participants with low levels of social anxiety for their study. If that is true (which we don’t know due to a lack of baseline assessment of IB), the negative CBM-I was indeed effective in reducing participants’ positive interpretation biases to a level that no significant difference was found between positive and negative interpretation biases in the negative group. Also, it would be the case that because participants already had more positive than negative IB, the positive CBM-I either did not impact their IBs or slightly increased them. Therefore, this cannot be concluded that the positive or negative CBM-I trainings were successful or not in changing participants’ interpretation biases as we don’t know participants’ initial baseline IB patterns. Although given that participants were randomly assigned to the CBM-I groups, it is very likely that their baseline IB were similar, and it can be concluded that their significant difference after the CBM-I training is due to the training, but still we don’t know if both groups had more positive than negative IB before the CBM-I training (in that case negative CBM-I was effective), or more negative than positive IB (in that case positive CBM-I was effective), or equal positive and negative IB.

Likewise, if authors hypothesize and expect that participants in the negative CBM-I show more negative than positive IB and those in the positive CBM-I show more positive than negative IB, they can say that these findings did not fully supported their hypothesis, but it is still incorrect to conclude that positive CBM-I was successful in changing IB and negative CBM-I failed to change participants IB for the same reason that I explained earlier.

The same problem can be seen in the discussion section too where the authors wrote:

“There was no effect of CBM-I training on interpretation bias in the negative group….”

They also write: “Second, the positive group revealed a positive interpretation bias in both the target and foil sentences, whereas the negative group did not show any difference before and after CBM training”. It is important to note that in this sentence, they mention “before and after CBM training,” but the interpretation bias assessment occurred only after the CBM-I training in the study.

I think this problem in the interpretation of the effects of the CBM-I training is very important and needs to be addressed throughout the manuscript.

Response. 

Thank you for your insightful comments and for bringing attention to the issue regarding the interpretation of the effects of the CBM-I training in our study. We appreciate your valuable feedback and have taken your points into careful consideration.

We acknowledge the limitation of our study in not conducting a baseline assessment of interpretation biases (IB) prior to the CBM-I training. As you rightly pointed out, the absence of baseline data makes it challenging to draw definitive conclusions about the effectiveness of the positive and negative CBM-I interventions. We have incorporated this limitation into our discussion, emphasizing the need for future research to include baseline assessments to provide a more comprehensive understanding of the impact of CBM-I interventions on IB.

On the other hand, it is important to note that participants in our study were randomly assigned to the positive and negative training groups, which helps minimize initial differences between the groups. Random assignment increases the likelihood of similarity in baseline IB patterns. Thus, while we acknowledge the limitations, the observed differences in IBs between the groups can be attributed to the training itself. Furthermore, it is crucial to clarify that our study focused on the effects of the CBM-I training on participants' interpretation biases within the specific context of the study session. We did not explicitly aim to assess long-term changes in interpretation biases. The reported results reflect the immediate impact of the CBM-I training, indicating that the positive training group showed a positive interpretation bias consistent with the trained valence, while the negative training group did not exhibit a significant effect of training on their interpretation biases. While we understand that this conclusion does not capture the entire complexity of participants' IB patterns, it provides valuable insights into the immediate impact of the CBM-I training within the study session.

Regarding the specific goals and hypotheses of our study, we agree that it is important to clarify that our study did not explicitly aim to demonstrate the success or failure of the interventions in changing IB. Our primary focus was on exploring the interaction between interpretation biases and memory biases in a single session. Therefore, the conclusions we drew were within the context of our specific research goals and hypotheses. In response to your feedback, we have made revisions to the manuscript to address the issue you raised regarding the comparison between pre-training and post-training IB assessments. We have explicitly stated that our assessment of IB occurred only after the CBM-I training and have avoided making any statements that imply a comparison between pre-training and post-training IB. We have also incorporated the limitation you suggested regarding the lack of baseline assessment of interpretation bias in the final version of the manuscript.

Overall, we appreciate your valuable feedback and believe that your suggestions have significantly enhanced the accuracy and clarity of our interpretations. We have carefully addressed the issues you raised and made the necessary revisions throughout the manuscript.

Thank you for your time and guidance in helping us strengthen our study.

P 22

Finally, we acknowledge that the absence of a baseline assessment of interpretation bias is a limitation of our study. Therefore, we urge caution in interpreting the results regarding the effect of CBM-I training on interpretation bias. Despite the random assignment of participants to the positive and negative training groups, we cannot disregard the possibility that there might have been initial differences in positive and negative interpretation tendencies between the groups. To gain a more comprehensive understanding of the impact of CBM-I training on interpretation biases, future research should consider incorporating baseline measurements. This would enable a clearer assessment of the effectiveness of CBM-I interventions in shaping interpretation biases.

Minor issues

1- Please specify the name of the biases in the first two sentences of the abstract i.e. interpretation and memory. Currently, the way the abstract starts talks about "each bias" and "each type of bias" which seems not clear and can include any type of different bias.

Response. Thank you for your helpful comments. We have revised the first two sentences of the abstract to provide clarity regarding the specific biases under consideration. The revised sentences now specify interpretation, attention, and memory biases. 

The combined effect of each cognitive bias, interpretation, attention, and memory bias, is known to play a causal role in the etiology and maintenance of social anxiety. However, little is known about how each type of bias (i.e., interpretation, memory bias) acts during social anxiety.

2- The order of two sentences in the introduction needs to be changed. Specifically, the sentence that starts with "Mathews and Mackintosh [10] found that participants in the positive CBM-I group...." seems to belong after its next sentence i.e. "Mathews and Mackintosh [10] argued that CBM-I, which induces positive interpretations in social anxiety, may have ..." not before it.

Response. Thank you for your valuable feedback. We have revised the order of the sentences in the introduction as per your suggestion.

---

## [Decision Letter · Decision Letter 2]

24 Jul 2023

Induced interpretation bias affects free recall and episodic memory bias in social anxiety

PONE-D-22-04709R2

Dear Dr. Lee,

We’re pleased to inform you that your manuscript has been judged scientifically suitable for publication and will be formally accepted for publication once it meets all outstanding technical requirements.

Kind regards,

Anindita Bhadra, PhD

Academic Editor

PLOS ONE

Additional Editor Comments (optional):

I am happy to accept the manuscript for publication. I am sorry that the reviewing process took very long, primarily due to the non availability of Reviewer 1.

Reviewers' comments:

Reviewer's Responses to Questions

**Comments to the Author**

1. If the authors have adequately addressed your comments raised in a previous round of review and you feel that this manuscript is now acceptable for publication, you may indicate that here to bypass the “Comments to the Author” section, enter your conflict of interest statement in the “Confidential to Editor” section, and submit your "Accept" recommendation.

Reviewer #1: All comments have been addressed

2. Is the manuscript technically sound, and do the data support the conclusions?

Reviewer #1: Yes

3. Has the statistical analysis been performed appropriately and rigorously? 

Reviewer #1: Yes

4. Have the authors made all data underlying the findings in their manuscript fully available?

Reviewer #1: Yes

5. Is the manuscript presented in an intelligible fashion and written in standard English?

Reviewer #1: Yes

6. Review Comments to the Author

Reviewer #1: The authors have addressed my concerns and I think the manuscript is now ready to be accepted. Congratulations!

7. PLOS authors have the option to publish the peer review history of their article (what does this mean?). If published, this will include your full peer review and any attached files.

Reviewer #1: No

---

## [Editor Report · Acceptance letter]

28 Jul 2023

PONE-D-22-04709R2 

Induced interpretation bias affects free recall and episodic memory bias in social anxiety 

Dear Dr. Lee:

I'm pleased to inform you that your manuscript has been deemed suitable for publication in PLOS ONE. Congratulations! Your manuscript is now with our production department. 

Kind regards, 

on behalf of

Dr. Anindita Bhadra 

Academic Editor

PLOS ONE